# Lung Cancer Screening: New Perspective and Challenges in Europe

**DOI:** 10.3390/cancers14092343

**Published:** 2022-05-09

**Authors:** Jan P. Van Meerbeeck, Emma O’Dowd, Brian Ward, Paul Van Schil, Annemiek Snoeckx

**Affiliations:** 1Department of Thoracic Oncology, Antwerp University and Antwerp University Hospital, 2650 Edegem, Belgium; jan.van.meerbeeck@uza.be; 2Nottingham University Hospitals NHS Trust, Nottingham NG7 2UH, UK; emma.odowd@nottingham.ac.uk; 3European Respiratory Society, Advocacy Department, 1000 Brussels, Belgium; brian.ward@ersnet.org; 4Department of Thoracic and Vascular Surgery, Antwerp University and Antwerp University Hospital, 2650 Edegem, Belgium; 5Department of Radiology, Antwerp University and Antwerp University Hospital, 2650 Edegem, Belgium; annemiek.snoeckx@uza.be

**Keywords:** lung cancer, screening, computed tomography, management

## Abstract

**Simple Summary:**

Screening for lung cancer in a high-risk population has been shown to be beneficial, with reduced mortality in large randomised trials. However, the general implementation of screening is not evident and many factors have to be considered. In this paper, we will review the current status of screening for lung cancer in Europe and the many hurdles that have to be overcome. Multidisciplinary cooperation between all specialists dealing with lung cancer is required to obtain the best outcome. Hopefully, Europe’s Beating Cancer Plan will incorporate screening for lung cancer to allow general implementation by similar programmes in every European Member State. This will also provide an opportunity for further, large-scale studies to refine the inclusion of specific risk populations, diagnosis and management of screen-detected nodules.

**Abstract:**

Randomized-controlled trials have shown clear evidence that lung cancer screening with low-dose CT in a high-risk population of current or former smokers can significantly reduce lung-cancer-specific mortality by an inversion of stage distribution at diagnosis. This paper will review areas in which there is good or emerging evidence and areas which still require investment, research or represent implementation challenges. The implementation of population-based lung cancer screening in Europe is variable and fragmented. A number of European countries seem be on the verge of implementing lung cancer screening, mainly through the implementation of studies or trials. The cost and capacity of CT scanners and radiologists are considered to be the main hurdles for future implementation. Actions by the European Commission, related to its published Europe’s Beating Cancer Plan and the proposal to update recommendations on cancer screening, could be an incentive to help speed up its implementation.

## 1. Introduction

There is clear evidence of benefit for screening with low-dose computed tomography (LDCT) in lung cancer. The National Lung Screening Trial (NLST) [1] and the Dutch–Belgian NELSON trial [2] have shown that LDCT reduces lung-cancer-specific mortality, and the NLST also showed a reduction in all-cause mortality of 6.7% [1]. Other smaller European trials, although underpowered, have also shown findings in keeping with the larger trials [3,4,5]. These trials provide enough evidence to support the implementation of screening for lung cancer based on clinical effectiveness, but there are still challenges that need to be overcome prior to the roll out of a population-based screening programme in Europe. This paper will review areas for which there is good or emerging evidence and the areas which still require investment, research or represent implementation challenges (Figure 1).

## 2. Recruitment, Eligibility Optimisation and Participation

To ensure that a programme is cost effective and to minimise harm, it is important to select the population most likely to benefit. Questions on how best to do this remain. Some patients may have symptoms related to a lung cancer [6]. Most trials used age and smoking-pack-year criteria to select participants. However, risk prediction models have been shown to be superior to selection using age and smoking status alone [7,8,9,10,11,12]. This may be due in part to these models incorporating more detailed smoking data and considering other risk factors such as chronic obstructive pulmonary disease (COPD) or asbestos exposure. The most widely used models are the Liverpool Lung Project version 2 (LLPv2) [13] and the Prostate, Lung, Colorectal and Ovarian Cancer Screening Trial model (modified 2012) (PLCOm2012) [9]. Currently, there is no consensus on which model to use, or even which threshold to use within a model. Many pilots and trials are using multiple models and include any participant who meets the predefined threshold of each model.

Advertising and questionnaires have been used as the initial method of recruitment in many trials to date, but the uptake of those eligible for screening is often not high. Furthermore, current recruitment models are very resource-intensive. The United Kingdom Lung Screening Trial (UKLS) used population primary care records to identify those aged 50–75 years and residing in specific regions, then asked them to complete and return a postal questionnaire. Thirty-one percent of eligible people responded to the questionnaire but only 11.5% of respondents were at a high enough risk for entry into the trial [14]. In a similar way, the NELSON trial used population registry data to identify those who were aged between 50 and 74 years and sent them brief information about the trial and a questionnaire to complete. Of 606,409 people approached, 25% returned the questionnaire and 21% of these met the trial eligibility criteria [2]. Other approaches have been tried in various UK pilots and trials, drawing on insight from screening programmes in similar populations; for example, invites from primary care and reminder/re-invitation letters [15,16].

It is important to understand how participants want to be contacted and the best ways to do so to maximize uptake in high-risk groups. The UKLS showed that those at highest risk were least likely to respond to an invitation to take up screening [11]. Work by Quaife et al. has shown that a targeted, stepped, and low-burden intervention invitation approach was able to better engage those living in areas of highest deprivation and lung cancer incidence, thus improving equity in screening uptake [17].

There has been concern about the low uptake of screening in the United States (US) since it was approved by the United States Preventive Services Task Force (USPSTF) in 2013. A recent study using data from the Lung Cancer Screening Registry, provided by the American College of Radiology, showed that only 2% of eligible, high-risk individuals were screened in 2016 [18]. Data from the Centers for Disease Control and Prevention’s 2017 Behavioral Risk Factor Surveillance System (BRFSS) survey also showed that uptake varied and was noticeably higher in those with health insurance compared with uninsured individuals (15.2% vs. 4.0%) [19]. So far, UK pilot and trial data have been much more favorable. The Lung Screen Uptake Trial (LSUT) performed in London had a 53% participation rate and 77% of those participants underwent LDCT screening [20]. Work is ongoing to examine the role of digital solutions and social media, both in recruitment and the assessment of eligibility, but there may need to be a bespoke/tailored approach in each country, depending on which data are already held on their population, to try to identify, invite and engage those eligible for screening.

## 3. Harm Minimization

To maximize the benefits from screening it is important that harm is minimized. One of the concerns from NLST was a high false positive rate. The use of evidence-based nodule management guidelines and protocols to clearly define a positive screen are important to reduce harm from over investigation and to improve cost effectiveness. The most readily used are the American College of Radiology (ACR) Lung RADS version 1.1 2019, available from: https://www.acr.org/-/media/ACR/Files/RADS/Lung-RADS/LungRADSAssessmentCategoriesv1-1.pdf?la=en (accessed on 26 April 2022), the British Thoracic Society (BTS) Pulmonary Nodule Guideline and the Fleischner Society Guidelines (for incidentally detected pulmonary nodules) [21,22]. Volumetric nodule assessment is preferred for screening; it has been shown to be more accurate than diameter measurement as it reduces variability between radiologists. Volumetric assessment is also more sensitive in the detection of early growth [23,24]. There were also concerns about the overdiagnosis rate in NLST, but long-term follow-up data at 10 years show that this was only 3.1% and was largely due to bronchoalveolar cell carcinoma [25]. Again, judicious management of nodules, particularly pure ground glass nodules, can help to reduce the proportion of participants who are over diagnosed and any associated harm.

NLST and NELSON had benign resection rates of 23% and 24%, respectively, with 61% of participants who did not have lung cancer having some form of diagnostic workup [26]. More recent pooled UK trial and pilot data on 11,148 participants show a much lower rate of harm, with only 0.6% of participants without lung cancer undergoing invasive testing and a benign resection rate of 4.6% [27]. This was largely achieved by adherence to nodule management protocols, malignancy risk assessment according to BTS guidelines and the careful use of percutaneous lung biopsy.

It is also important to minimize radiation exposure. Low-dose CT has reduced the risk associated with ionizing radiation. A study using screening data from Italy estimated that there would be one radiation-induced major cancer for every 108 lung cancers detected through screening, with a median cumulative effective dose of CT screening of roughly 9 millisievert (mSv) for men and 13 mSv for women, after 10 years of screening [28]. In the future, advances in technology will certainly allow the further reduction of this risk.

## 4. Cost Effectiveness and Add-On Health Interventions

Recent cost effectiveness estimates from multiple sources have suggested that LDCT screening can be cost effective depending on the threshold set by each country [29,30,31,32,33,34,35,36]. The integration of smoking cessation is a key aspect of any lung cancer screening programme and has been shown to substantially improve cost effectiveness, although the optimal strategy for integrating this is still unclear and the focus of ongoing research [33,37]. Personalised risk stratification following a baseline CT may improve cost effectiveness further. In the NELSON study, those with a negative baseline scan had a 3% chance of developing lung cancer at 10 years, compared with a 52% chance in those with a positive screen [2]. It is not yet known what the potential added health benefit may be of identifying and treating other conditions that may be picked up as part of LDCT screening, such as chronic obstructive pulmonary disease (COPD) and coronary artery disease (CAD). The 4-IN-THE-LUNG-RUN trial is a multicenter implementation trial underway in five European countries and is aiming to answer questions about personalized strategies in recruitment, screening intervals (biennial versus annual in those with a negative baseline LDCT), smoking cessation and other add-on health interventions for co-morbidity such as CAD and COPD [38].

## 5. Incidental Findings

Incidental findings are common in CT screening, but the over investigation and over reporting of these can cause anxiety, lead to harm from unnecessary tests and increased costs. The NELSON group reviewed a subset of their participants and showed that 73% had non-clinically relevant incidental findings [39]. Data from the UK have shown that by using clear guidelines and protocols for the management of incidental findings, those requiring further investigation or onward referral are actually very low [40,41].

## 6. Workforce/Capacity

Across Europe there is a huge shortage in trained radiologists, with the UK having the lowest number of practicing radiologists at 4.7 per 100,000 of the population [42]. Radiographer numbers are also of concern. The Diagnostic Radiography Workforce UK Census 2020 showed an average vacancy rate from respondents of 10.5% [43]. There is hope that artificial intelligence (AI) solutions may help to mitigate some of these concerns, but questions remain on how and where it is best to deploy these tools in clinical practice, and further research is required to optimise their performance. These workforce challenges are not unique to radiology and will affect all diagnostic and treatment specialists. A study by Rodin at al. highlighted inequities in access to radiotherapy machines, radiation oncologists and medical physicists across Europe [44]. Access to CT scanners also varies widely between countries. Data on the number of CT scanners by country and per million of the population are presented in Figure 2 and show the wide geographical variation in availability [45]. There is a real need for investment in infrastructure, more state-of-the-art technology and attention given to workforce planning/capacity assessments on a country-by-country basis.

Thoracic surgeons also play a major role in lung cancer screening, especially regarding decisions on further invasive investigations for screen-detected nodules [46]. In 2017, the European Society of Thoracic Surgeons (ESTS) created a working group to discuss specific surgical aspects. Besides the implementation of CT screening in Europe, the following topics were covered:Participation of thoracic surgeons in CT screening programmes;Training and clinical profile for surgeons participating in CT screening programmes;The use of minimally invasive thoracic surgery and other relevant surgical issues;Associated elements such as smoking cessation, nodule evaluation algorithms and pathology reports.

## 7. Quality Assurance

For any screening programme, a key factor in its success is a standardised evidence-based approach and robust quality assurance (QA). The Targeted Lung Health Check Programme in England has produced a Quality Assurance Standards document and a Standard Protocol which covers the specifics of governance, technical standards for hardware and software, and the management of all downstream outcomes, such as the communication of results, clinical pathways and the management of incidental findings [47,48]. There are also two European Respiratory Society (ERS) Taskforces underway aiming to produce pan-European consensus documents for a technical standard in LDCT screening and a statement on the management of incidental findings.

## 8. The European Union (EU) and Lung Cancer Screening

In the 1980s, French President François Mitterrand and other EU heads of government took an interest in fighting the scourge of cancer. Since then, the EU has put forward multiple initiatives to tackle cancer, despite the EU’s shared or limited competences in health (https://www.eumonitor.eu/9353000/1/j9vvik7m1c3gyxp/vh7bhpj5azzb) (accessed on 26 April 2022). At the start of the 21st century, one of these initiatives was the recommendation of the European Union (EU) Council that urged Member States, in 2003, to devote greater attention to breast, cervical and colorectal cancer screening, as there was established evidence that the implementation of organised screening through a population-based programme could significantly reduce mortality from these cancers.

Council recommendations are influential, set standards and often have EU funding linked to them to assist their implementation. While they raise political expectations and government commitments, they are not legally binding [49]. As a result, there is greater heterogeneity in the approaches of the Member States to organise quality-assured services in the context of population-based cancer screening programmes, than would be the case with a legally binding regulation. This is reflected in the wide variations in the invitation coverage and the suboptimal achievements of some Member States in the decade after the Council recommendations were announced [50]. Furthermore, it appeared that converting opportunistic testing to a population-based organised screening was much more challenging than launching the programme as a population-based one from the beginning. Hence, the Council’s recommendations require updating, including making them more stringent, explicit and inclusive.

One of the many actions of Europe’s Beating Cancer Plan, published by the European Commission in 2021, proposes to update the Council recommendation on cancer screening to ensure it reflects the latest available scientific evidence, and to consider extending targeted cancer screening to include additional cancers, such as lung, prostate and gastric cancer (Europe’s Beating Cancer Plan. Brussels: European Commission 2021). This is a priority action of the Cancer Plan and the only one to be specifically mentioned in the state of the union letter of intent for 2022, by Commission President von der Leyen (https://www.europarl.europa.eu/legislative-train/theme-promoting-our-european-way-of-life/file-cancer-screening, accessed on 26 April 2022). The EU commission will be informed by advice from the Group of Chief Scientific Advisors, prepared by early 2022 at the latest (Figure 3)

The assessment of the inclusion of other cancers will be carried out according to the criteria of the Council Recommendation on cancer screening:Implement new cancer screening tests in routine healthcare only after they have been evaluated in randomised controlled trials;Run trials, in addition to those on screening-specific parameters and mortality, on subsequent treatment procedures, clinical outcomes, side effects, morbidity and quality of life;Assess the level of evidence concerning the effects of new methods by pooling trial results from representative settings;Consider the introduction into routine healthcare of potentially promising new screening tests, which are currently being evaluated in randomised controlled trials, once the evidence is conclusive and other relevant aspects, such as cost effectiveness in the different healthcare systems, have been taken into account.

In its report entitled Improving Cancer Screening in the European Union, SAPEA issued a strong recommendation for extending screening programmes to lung cancer screening by low-dose CT scanning, based on effectiveness and mortality burden (www.sapea.info/cancer-screening, accessed on 26 April 2022).

In parallel, the European Parliament set up a Special Committee on Beating Cancer (BECA) to deliver its position on the Cancer Plan. The BECA commission calls for recognising the evidence that proves the positive effect of targeted lung cancer screening on mortality and encourages the Council, based on the outcome of the above-mentioned assessment, to consider including lung and prostate cancer screening in the update of the Council recommendations in 2022 (https://www.europarl.europa.eu/doceo/document/TA-9-2022-0038_EN.pdf, accessed on 26 April 2022).

Thirdly, a public consultation on the update of the EU cancer screening recommendations will soon be launched by the Commission. The vote in Parliament on their report is planned for late 2022.

Several health organisations, such as the European Respiratory Society (ERS), European Society of Radiology, European Cancer Patient Coalition (ECPC), Lung Cancer Europe (LuCE), the Irish Cancer Society and many others, successfully advocated for a number of amendments to the European Parliament report that, for example, set clear and tangible targets for Member States to implement lung cancer screening (https://www.ersnet.org/wp-content/uploads/2021/10/Open-letter_ERS_Updated-13.10.2021.pdf) (accessed on 26 April 2022):By 2024 the European Commission should develop and publish new guidelines on lung cancer screening for high-risk groups;By 2026 at least five EU Member States should have incorporated these guidelines into their national cancer plans;By 2027, all EU Member States should have put in place a strategy for the early detection of lung cancer in the high-risk population.

In the meantime, objection to these amendments and other text unfavourable to screening comes from various scientific and political entities inside and outside of the EU, claiming a lack of evidence, the need for cost effectiveness studies and the prior adoption by the World Health Organization (WHO). Continued opposition to lung cancer screening from certain quarters is expected even if the chief scientific advisors issue a favourable opinion and it is therefore hard to predict if lung cancer screening will be included in the final updated Council recommendation expected to be presented by the Commission in the third quarter of 2022.

## 9. Implementation of Lung Cancer Screening in Europe

The picture in early 2022 for lung cancer screening in Europe is heterogenous and its implementation is variable. At the time of writing, only two countries have implemented population-based lung cancer screening programmes in a high-risk population: Croatia, as of October 2020 https://www.euapm.eu/pdf/EAPM_revolutionising_lung_cancer_care_all_together_croatia_feb_2021.pdf (accessed on 26 April 2022) and Poland as of 2021 (population-based pilot) [51]. Other governments cite the lack of a Council recommendation as a reason not to implement screening. Finally, several countries are at a various stages of implementation. To obtain a current update, in January 2022 we conducted an email survey among key opinion leaders (KOLs) in European countries, with seven specific questions (Table 1). Replies were obtained from 23 countries: Austria, Belgium, Bulgaria, Czech Republic, Denmark, Estonia, France, Germany, Greece, Hungary, Iceland, Italy, Lithuania, Luxembourg, Norway, Poland, Portugal, Serbia, Slovakia, Spain, Switzerland, the Netherlands and the United Kingdom.

Out of those who responded, only the Czech Republic has an ongoing population-based nationwide lung cancer screening programme. In total, 3 out of 22 countries have plans to start population-based screening in 2022. Only three countries require formal EU recommendation before embarking on lung cancer screening. The implementation and regulatory policy is well defined in 57.1% of countries. In 47.8% of countries, institutions are currently participating in national or international large-scale feasibility projects or implementation pilot studies. Of these countries, in 7 out of 11 there is a national/regional lung cancer screening protocol or task force overseeing lung cancer screening. Of the 12 countries without current lung cancer screening pilot trials, only two are planning to start trials in the upcoming year. A cost effectiveness study has been published in less than a quarter of countries (5/23). In half of the countries where this has not been yet carried out, this cost effectiveness analysis is compulsory for population-based lung cancer screening to be implemented. In 68% (16/23) of countries, the costs of a national population-based lung cancer screening programme will be covered by health authorities and/or national health services. In only in a limited number of countries (4/23) is this covered by insurance companies.

Decisions regarding implementation take place at the national level (health minister) in 82.7% (19/23) of countries. Cost and capacity of CT-scans and radiologists are considered the main hurdles for future implementation, rather than the stigma of lung cancer or the willingness of radiologists.

The inclusion of lung cancer screening in the update of the Council recommendation would be a major step that we truly hope is made. It would be a strong signal that the EU’s cancer plan can make a meaningful difference, by advancing a measure already shown to save lives from the deadliest of cancers (https://www.sts.org/publications/news-surgeons-view/lung-cancer-screening-saves-lives) (accessed on 26 April 2022).

## 10. Conclusions

It is clear that LDCT screening for lung cancer reduces lung cancer mortality. Some countries are starting to make progress with implementing screening programmes, but protocols and quality assurance are required to both maximise the benefit and minimise harm. Implementation research and demonstration programmes examining optimal selection and recruitment strategies, the management of incidental findings and the integration of smoking cessation need to be prioritised. Workforce and CT capacity is a potential barrier in some countries and investment in infrastructure and forward planning, alongside the integration of AI solutions, is also key to mitigate these. It is expected that population-based lung cancer screening will be fragmentally and variably implemented throughout Europe in the following 4–5 years.

## Figures and Tables

**Figure 1 cancers-14-02343-f001:**
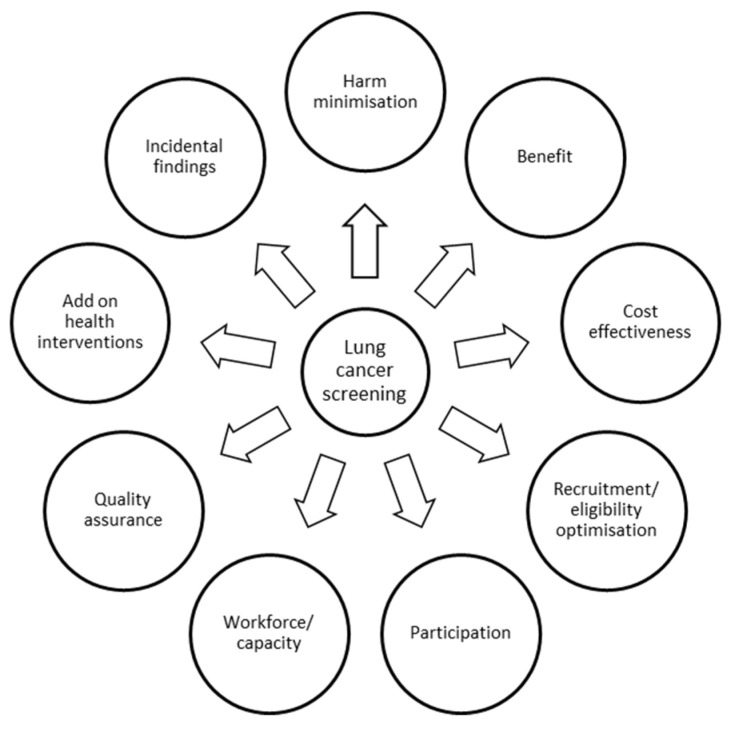
Summary of areas to be addressed prior to lung cancer screening implementation.

**Figure 2 cancers-14-02343-f002:**
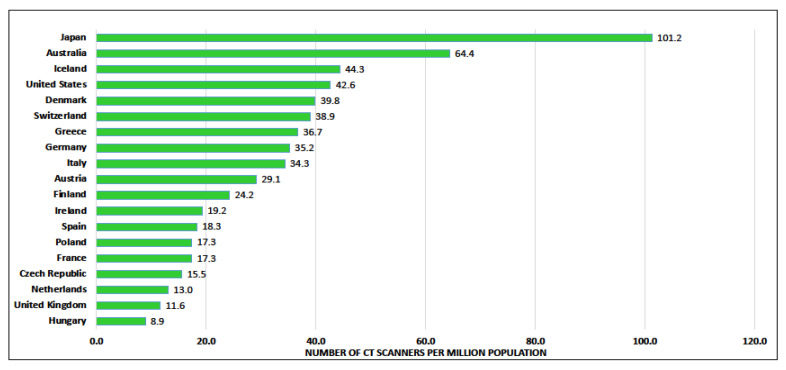
Number of CT scanners per million inhabitants.

**Figure 3 cancers-14-02343-f003:**
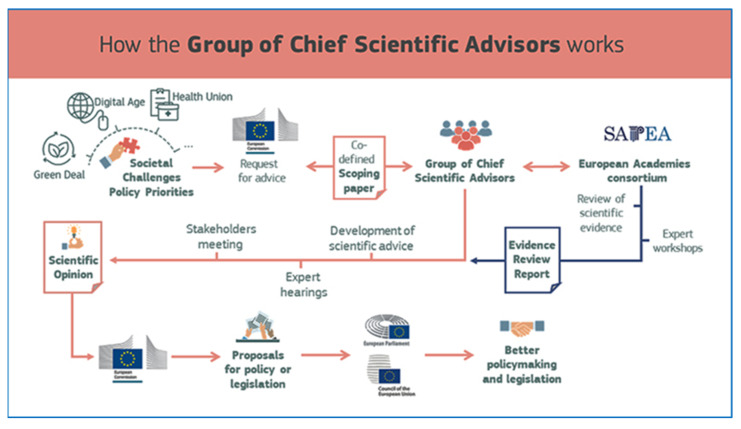
The Federation of European Academies of Medicine (FEAM) collects, presents and reviews the evidence for the Commission’s Group of Chief Advisors (SAPEA). SAPEA will then draft a proposal of recommendations to the European Commission.

**Table 1 cancers-14-02343-t001:** Survey of key opinion leaders from European countries: questions and results.

**Question 1: Is there as of 1 January 2022 an ONGOING population-based nationwide lung cancer screening (LCS) programme with low dose CT scan in your country?**
Yes: 1/23 (4.3%) No: 22/23 (95.7%) → If NO, will there be one starting in 2022? Yes: 3/22 (13.6%) No: 19/22 (86.4%) → If NO: does your country require a formal EU-recommendation before embarking on lung cancer screening? Yes: 3/22 (13.6%) No: 19/22 (86.4%) → If NO: is the policy and regulatory process governing the decision to implement nation-wide screening programmes well defined in your country? Yes: 12/21 (57.1%) No: 9/21 (42.9%)
**Question 2: Are institutions in your country currently participating in feasibility projects, piloting the implementation of LCS with low dose CT scan in your country?**
Yes: 11/23 (47.8%) → If YES, is there a national screening protocol or task force? Yes: 7/11 (63.6%) No: 4/11 (36.4%) No: 12/23 (52,2%) → If NO: will there be one starting in 2022? Yes: 2/12 No: 8/12 No answer/unclear: 2/12
**Question 3: Has any cost-effectiveness study been published regarding LCS in your country?**
Yes: 5/23 (21.7%) No: 18/23 (78.3%) → If NO: is a cost-effectiveness study required for lung cancer screening to be implemented? Yes: 9/18 (50%) No: 9/18 (50%)
**Question 4: Who will cover the costs of a national population based lung cancer screening programme in your country?**
Health authority and/or National health service: 16/23 (69.6%) Insurance companies: 4/23 (17.4%) Other: 3/23 (13.0%)
**Question 5: Who decides on the implementation of a population based LCS in your country?**
National Health minister or Health board: 19/23 (82.7%) Board of directors of health insurance companies: 3/23 (13.0%) Regional or local health authority: 2/23 (8.7%) Other: 3/23 (13.0%)
**Question 6: Will the implementation of lung cancer screening be conditional of a structured smoking cessation intervention?**
Yes: 9/23 (39.1%) Very likely: 8/23 (34.8%) Likely: 4/23 (17.4%) Unlikely: 2/23 (8.7%) No: 0/23
**Question 7: What are according to you the main hurdles for the implementation of a population-based national lung cancer screening programme in your country? Ranking of cost–public opinion on stigma of lung cancer–capacity of CT-scans and radiologists–willingness of GP’s/radiologists-others**
Most important hurdle: Cost 10/23 (43.5%) Public opinion on stigma of lung cancer 3/23 (13.0%) Capacity of CT-scans and radiologists 5/23 (21.7%) Willingness of general practitioners’s/radiologists 2/23 (8.7%) Other 3/23 (13.0%) Second most important hurdle: Cost 9/23 (39.1%) Public opinion on stigma of lung cancer 4/23 (17.4%) Capacity of CT-scans and radiologists 6/23 (26.1%) Willingness of general practitioners’s/radiologists 3/23 (13.0%) Other 1/23 (4.3%) Third most important hurdle: Cost 4/23 (17.4%) Public opinion on stigma of lung cancer 0/23 Capacity of CT-scans and radiologists 5/23 (21.7%) Willingness of general practitioners’s/radiologists 7/23 (30.4%) Other 7/23 (30.4%)

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
