# Peer review of "Lung Cancer Screening: New Perspective and Challenges in Europe"

_cancers, 2022, doi:10.3390/cancers14092343_

Round 1

Reviewer 1 Report

Title: could be changed in Lung cancer screening new perspective and challenges in Europe: state of the art.

Introduction

  • line 60, In 2011, the U.S. National Lung Cancer Screening Trial (NLST) reported a 20% reduction of lung cancer mortality after regular screening by low-dose computed tomography (LDCT), as compared to X-ray screening. Lung cancer screening programs in Europe await confirmation of these first findings from European trials that started in parallel with the NLST. The German Lung cancer Screening Intervention (LUSI) is a randomized trial among 4,052 long-term smokers, 50-69 years of age, recruited from the general population, comparing five annual rounds of LDCT screening (screening arm; n = 2,029 participants) with a control arm (n = 2,023) followed by annual postal questionnaire inquiries. Data on lung cancer incidence and mortality and vital status were collected from hospitals or office-based physicians, cancer registries, population registers and health offices. Over an average observation time of 8.8 years after randomization, the hazard ratio for lung cancer mortality was 0.74 (95% CI: 0.46-1.19; p = 0.21) among men and women combined. Modeling by sex, however showed a statistically significant reduction in lung cancer mortality among women (HR = 0.31 [95% CI: 0.10-0.96], p = 0.04), but not among men (HR = 0.94 [95% CI: 0.54-1.61], p = 0.81) screened by LDCT (pheterogeneity = 0.09). Findings from LUSI are in line with those from other trials, including NLST, that suggest a stronger reduction of lung cancer mortality after LDCT screening among women as compared to men. This heterogeneity could be the result of different relative counts of lung tumor subtypes occurring in men and women. please discuss and cite doi:10.1002/ijc.32486.
  • line 114, however, the clinic must be considered for its role in diagnosis through symptoms that are related to a high risk of pulmonary neoplasia such as involvement of surrounding vascular or nerve mediastinal structures. please discuss and cite doi:10.1016/j.mayocp.2019.01.013. and  doi:10.14639/0392-100X-680
  • line 133, Targeted therapy could be an option for the treatment of lung cancer. Evaluation of the cost-effectiveness of computerized tomographic colonography (CTC) in lung cancer screening is recommended. The perspective of the community should be taken into consideration in studies of cost-effectiveness. Paying more attention to the topic of discounting will be necessary in the studies. please discuss and cite doi:10.1186/s12913-017-2374-1

Author Response

Reviewer 1

Title: could be changed in Lung cancer screening new perspective and challenges in Europe: state of the art.

Response: we thank the reviewer for this suggestion and changed the title into “Lung cancer screening: new perspective and challenges in Europe”. We didn’t mention “state of the art” as our paper cannot really be considered a state of the art paper.

Introduction

  • line 60, In 2011, the U.S. National Lung Cancer Screening Trial (NLST) reported a 20% reduction of lung cancer mortality after regular screening by low-dose computed tomography (LDCT), as compared to X-ray screening. Lung cancer screening programs in Europe await confirmation of these first findings from European trials that started in parallel with the NLST. The German Lung cancer Screening Intervention (LUSI) is a randomized trial among 4,052 long-term smokers, 50-69 years of age, recruited from the general population, comparing five annual rounds of LDCT screening (screening arm; n = 2,029 participants) with a control arm (n = 2,023) followed by annual postal questionnaire inquiries. Data on lung cancer incidence and mortality and vital status were collected from hospitals or office-based physicians, cancer registries, population registers and health offices. Over an average observation time of 8.8 years after randomization, the hazard ratio for lung cancer mortality was 0.74 (95% CI: 0.46-1.19; p = 0.21) among men and women combined. Modeling by sex, however showed a statistically significant reduction in lung cancer mortality among women (HR = 0.31 [95% CI: 0.10-0.96], p = 0.04), but not among men (HR = 0.94 [95% CI: 0.54-1.61], p = 0.81) screened by LDCT (pheterogeneity = 0.09). Findings from LUSI are in line with those from other trials, including NLST, that suggest a stronger reduction of lung cancer mortality after LDCT screening among women as compared to men. This heterogeneity could be the result of different relative counts of lung tumor subtypes occurring in men and women. please discuss and cite doi:10.1002/ijc.32486.

Response: we agree with the reviewer that there are other trials besides NLST and Nelson investigating CT scanning for lung cancer screening. We added following sentence on lines 51-53 of the marked revised manuscript: “Other smaller European trials, although underpowered, have also shown findings in keeping with the larger trials [3-5].” Three additional references were added:

  1. Becker N, Motsch E, Trotter A, Heussel CP, Dienemann H, Schnabel PA, et al. Lung cancer mortality reduction by LDCT screening-Results from the randomized German LUSI trial. Int J Cancer. 2020;146(6):1503-13.
  2. Pastorino U, Silva M, Sestini S, Sabia F, Boeri M, Cantarutti A, et al. Prolonged lung cancer screening reduced 10-year mortality in the MILD trial: new confirmation of lung cancer screening efficacy. Ann Oncol. 2019;30(7):1162-9.
  3. Field JK, Vulkan D, Davies MPA, Baldwin DR, Brain KE, Devaraj A, et al. Lung cancer mortality reduction by LDCT screening: UKLS randomised trial results and international meta-analysis. Lancet Reg Health Eur. 2021;10:100179.

We didn’t want to provide too many details on these trials as they are well known and discussed in detail in their respective papers and additional editorials.

  • line 114, however, the clinic must be considered for its role in diagnosis through symptoms that are related to a high risk of pulmonary neoplasia such as involvement of surrounding vascular or nerve mediastinal structures. please discuss and cite doi:10.1016/j.mayocp.2019.01.013. and  doi:10.14639/0392-100X-680

Response: We agree that consideration of specific symptoms may be important and following sentence was added on line 64 in the marked revised manuscript: “Some patients may have symptoms related to a lung cancer [6].” Reference 6 was added as suggested by the reviewer:

  1. Duma N, Santana-Davila R, Molina JR. Non-Small Cell Lung Cancer: Epidemiology, Screening, Diagnosis, and Treatment. Mayo Clin Proc. 2019;94(8):1623-40.

The suggested reference on voice rehabilitation is beyond the scope of our manuscript and was not included.

  • line 133, Targeted therapy could be an option for the treatment of lung cancer. Evaluation of the cost-effectiveness of computerized tomographic colonography (CTC) in lung cancer screening is recommended. The perspective of the community should be taken into consideration in studies of cost-effectiveness. Paying more attention to the topic of discounting will be necessary in the studies. please discuss and cite doi:10.1186/s12913-017-2374-1

Response: Our paper is about screening for lung cancer with LDCT rather than the cost effectiveness of CT-colonography (CTC) for colon cancer and possible extension to lung cancer screening. Adding a discussion on CTC could be confusing to the readership as we wanted to limit our paper to LDCT screening for lung cancer. There are no trials or cost effectiveness analyses yet to look at the role of CTC in lung cancer detection or the cost effectiveness of this approach so, although this paper poses an interesting question, we do not think it has relevance to the cost effectiveness of LDCT screening as discussed in this paper. Also, specific treatment with targeted therapies for screen-detected nodules is beyond the scope of our manuscript as it has not been thoroughly evaluated in this setting, but may be considered in future trials.

Additional response: the English language has been checked by a native English speaker (Emma O’Dowd) who is co-author and native English speaker.

Reviewer 2 Report

Dear Authors, thank you so much for submitting the manuscript titled “Implementation of lung cancer screening in Europe: status and challenges in 2022” to /Cancers/ to raising the titled issue (2022).  This is a long-term issue.

The scope of /Cancers/ is “/Cancers/ publish  articles including basic, translational, and clinical studies on all tumor types”. The context of the current review manuscript is out of the scope of /Cancers/. It is more suitable to be published in a newspaper or some other social media etc. to attract all the countries’ attention in Europe or even global. It’s more likely and close to a social issue. Hope the governments and committees of Europe countries could work together and try to solve the problems that the authors raising here. Despite this, the followings are some concerns and comments have been pointed out that the authors may want to consider.

Concerns and Comments:

  1. Line 7: The authors’ institute serial number “1” is missing.
  2. Line 37, Keywords: No keyword “diagnosis” appears in the main context. Only one “diagnostic” on line 116. No keyword “lung nodules” appears in the main context as well. They are not suitable to be listed as keywords at all.
  3. Almost all of the citations are not in their proper position. Please check throughout the whole manuscript. The citation should be inside of the sentence.
  4. There are lots of extra spaces throughout the manuscript. Please check.
  5. Line 80: There is an extra “)” before the last word “work”.
  6. Line 91: There is an extra “)” before the word “UK”.
  7. Lines 291-305: I’d suggest the authors make a summary table for this paragraph for clearer tracking.

Author Response

Reviewer 2

Dear Authors, thank you so much for submitting the manuscript titled “Implementation of lung cancer screening in Europe: status and challenges in 2022” to /Cancers/ to raising the titled issue (2022).  This is a long-term issue.

The scope of /Cancers/ is “/Cancers/ publish  articles including basic, translational, and clinical studies on all tumor types”. The context of the current review manuscript is out of the scope of /Cancers/. It is more suitable to be published in a newspaper or some other social media etc. to attract all the countries’ attention in Europe or even global. It’s more likely and close to a social issue. Hope the governments and committees of Europe countries could work together and try to solve the problems that the authors raising here. Despite this, the followings are some concerns and comments have been pointed out that the authors may want to consider.

Response: our paper belongs to a special section “Cancer therapy” with a special issue on “Lung adenocarcinoma: screening and surgical treatment”. So, screening is incorporated in the title of this special issue.  In fact, screening is a timely subject which is currently heavily discussed on a national and European level. Our manuscript provides an update on the current situation in Europe and in this way, may contribute to a better understanding and more general implementation of lung cancer screening in Europe. in this way, it falls within the scope of the journal “Cancers”.

Concerns and Comments:

  1. Line 7: The authors’ institute serial number “1” is missing.

Response: this has been corrected

  1. Line 37, Keywords: No keyword “diagnosis” appears in the main context. Only one “diagnostic” on line 116. No keyword “lung nodules” appears in the main context as well. They are not suitable to be listed as keywords at all.

Response: keywords lung nodules and diagnosis have been taken out (line 40 in revised, marked manuscript)

  1. Almost all of the citations are not in their proper position. Please check throughout the whole manuscript. The citation should be inside of the sentence.

Response: this has been corrected throughout the manuscript and numbering of references has been adapted after introduction of new references

  1. There are lots of extra spaces throughout the manuscript. Please check.

Response : in the Word file provided by the editors, central alignment is used which is responsible for some extra spaces to nicely fit the words within the space provided

  1. Line 80: There is an extra “)” before the last word “work”.

Response: this has been corrected

  1. Line 91: There is an extra “)” before the word “UK”.

Response: this has been corrected

  1. Lines 291-305: I’d suggest the authors make a summary table for this paragraph for clearer tracking.

Response:  table 1 has been expanded and responses to the specific survey questions were incorporated inside the table

Additional response: the English language has been checked by a native English speaker (Emma O’Dowd) who is co-author and native English speaker.

Round 2

Reviewer 2 Report

With respect, I seriously reviewed and checked the manuscript several times. My second-round review comments are as follows:

  1. Lines 51-53: If the “smaller European trials” indicates references 3-5 compared with “larger trials” references 1-2. I’d suggest the authors cited “[3-5]” followed “smaller European trials”, and “[1-2]” followed “larger trials” to avoid misunderstanding.
  2. Lines 56-57: I’d suggest the authors emphasize “lung cancer screening” in this sentence as it is the goal of this review.
  3. Line 60 Figure 1: Provide higher resolution Figure 1 as it is hard to read.
  4. Figure 1 and different sections in the review: I’d suggest the authors add a serial number in Figure 1 summarized areas to match the order of review sections in the manuscript for easier tracking by readers.
  5. Line 64: I’d suggest to the authors it might be better to use “Symptoms related to a type of lung cancer” or “Symptoms related to lung cancer” instead of “Symptoms related to a lung cancer”.
  6. Line 64: “Most trials used…”; Line 72: “Many pilots and trials are using…”. The tense seems confusing in this paragraph.
  7. Line 79: “per cent” should be “percent”.
  8. Lines 79-80: Please homogenous “Thirty-one percent”, “11.5%”.
  9. Line 82: “Between 50 and 74 years” should be “Between 50 and 74 years old”.
  10. Line 107 section 3. Harm minimization: I’d suggest the authors generate a table to summarize this paragraph for easier tracking and comparing different trials in different countries.
  11. Line 139: The authors focused on the lung cancer screening issues in Europe to discuss the cost effectiveness and add on health interventions in section 4. What’s the prospers of reference 36 /Yuan J, Sun Y, Wang K, Wang Z, Li D, Fan M, et al. Cost Effectiveness of Lung Cancer Screening With Low-Dose CT in Heavy Smokers in China. Cancer Prev Res (Phila). 2021./, from China. Please state and add descriptions if the authors wanted to compare the Countries in Europe with other non-Europe Countries.
  12. Line 142: Did the authors realize the reference 33 /Cressman S, Peacock SJ, Tammemagi MC, Evans WK, Leighl NB, Goffin JR, et al. The Cost-Effectiveness of High-Risk Lung Cancer Screening and Drivers of Program Efficiency. J Thorac Oncol. 2017;12(8):1210-22./ was a study from Canadian? Please state and add descriptions if the authors wanted to compare the Countries in Europe with other non-Europe Countries. Please confirm this study is still ongoing. Are you sure?
  13. Line 178 Figure 2: Provide higher resolution Figure 2 as it is hard to read.
  14. Lines 182-188: I’d suggest the authors summarize highlights of each topic instead of just listing titles in the manuscript.
  15. Provide higher resolution Figure 3.
  16. Line 256: There is an extra “]” followed by the word “plan”.
  17. Line 262: Where are the other two parts before “Thirdly”? It’s hard to track.
  18. Line 324: Homogenous the format of numbers “ Three out of 22”.
  19. Line 348 Conclusions section: I’d highly suggest the authors summarize a table before the conclusion part, which might be in the introduction part, to show that LDCT screening reduces lung cancer mortality from real-world clinical trials to emphasize the screening are very important.
  20. Lines 461-462: I can’t target where the useful information from the link provided in reference 42 is. The link goes to the homepage of “TMC a Unilabs company”. Did the authors mean the readers have to search the website to find it themselves? Please double-check all the references and links no matter where this manuscript will be published.
  21. Line 462: The right part “]” is missing.
  22. Line 468: I can’t find reference 45. Please provide detailed information. Please double-check all the references.